## [Transparent Peer Review file · Nature Communications]

Structure and mechanism of the HECT ligase HECTD3

Corresponding Author: Dr Katrin Rittinger

Version 0:

Reviewer comments:

Reviewer #1

(Remarks to the Author)

This manuscript by Huber et al. presents a comprehensive biochemical and structural analysis of HECTD3, a member of the HECT E3 ligase family. While HECT ligases are central regulators of ubiquitin-mediated signaling, their mechanisms of substrate selection and chain specificity remain incompletely understood. HECTD3 has been implicated in inflammation, apoptosis, infection, and cancer progression, yet its structure and enzymatic activity have remained largely uncharacterized. The authors combine biochemical ubiquitination assays, E2 profiling, linkage type analysis, cryo-EM and crystallography, SAXS, XL-MS, and mutational studies to provide the first full-length structural snapshots of HECTD3 in both apo and ubiquitin-loaded states. They reveal a novel N-terminal fold, identify unusual E2 preferences (favoring UBE2D family members over UBE2L3), and present evidence suggesting that HECTD3 catalyzes mainly K11- and K48-linked ubiquitin chains. The study further validates PARP1 as a substrate, maps ubiquitination sites, and proposes mechanistic roles for specific residues in chain-type determination and substrate binding.

The manuscript delivers a rigorous and technically impressive characterization of HECTD3, combining structural biology, biochemistry, and substrate validation to provide new insights into this poorly understood ligase. The cryo-EM reconstructions of full-length HECTD3, supported by crystallography of the DOC domain and SAXS data, are of particular value, revealing two unique N-terminal folds (D3NA and D3NB) and conformational flexibility between L- and inverted T-states characteristic of HECT ligases. The structural work is highly convincing and, aside from a minor figure presentation issue, excellent. The biochemical analyses, however, raise concerns that require further clarification and more cautious interpretation.

1) PARP1 binding and ubiquitination

The XL-MS data supporting PARP1 interaction with HECTD3 are strong, but the biochemical validation is less convincing. In Figure 6a, the Coomassie-stained SDS-PAGE shows only a modest shift in PARP1 molecular weight, suggesting attachment of few ubiquitin molecules, whereas the western blot in Supplementary Figure S10c shows distinct higher-molecular-weight species. It would be helpful if the authors clarified whether PARP1 is mainly multi-mono-ubiquitinated or polyubiquitinated. Do the DOC domain mutants (Y252A, F256A) abolish this higher-molecular-weight species in western blotting, and do they also impair HECTD3 auto-ubiquitination? Furthermore, can the authors exclude that these mutations interfere with E2–E3 charging? Finally, is the unusual preference for UBE2D E2s over UBE2L3 also observed in PARP1 ubiquitination assays?

2) Reported substrates not ubiquitinated in vitro

The failure to ubiquitinate caspase-8, caspase-9, or STAT3 in vitro is surprising, since these proteins are reported in vivo substrates. Could this reflect a requirement for post-translational modifications or adaptor proteins absent in the assay system? For example, phosphorylation of HECTD3 at Thr157 has been reported as essential for caspase-9 ubiquitination, and MALT1 could act as an adaptor for STAT3 or caspase-8. In addition, Supplementary Figure 10 shows that HECTD3 auto-ubiquitination is absent in the presence of caspase-8 and caspase-9, which is puzzling since both are apparently no suitable substrates competing with the former activity; an observation that requires further explanation. The authors should add a statement to the discussion addressing why the majority of reported substrates were not modified under their assay conditions. Ultimately, the idea that a constitutively active, and thus uncontrolled, E3 ligase is present in the cytosol is to some extent worrisome. Is it possible that an additional layer of regulation that restricts HECTD3 from futile or even detrimental ubiquitination events is present in vivo. Although a verification of such regulatory mechanisms is most likely beyond the scope of this study, the authors might still want to include some of their thoughts in the discussion.

3) Ubiquitin chain type

The conclusion that HECTD3 primarily catalyzes K11- and K48-linked chains is not fully convincing. Band patterns in Figure 1b and 1c suggest attachment of only a few ubiquitin molecules rather than chain elongation. In UbiCRest assays, residual ubiquitin signals after Cezanne or OTUB1 treatment could reflect priming ubiquitins rather than minor alternative linkages as the authors suggest. Moreover, OTUB1 treatment generates a strong, apparently monoubiquitinated species that persists even with Cezanne co-treatment, possibly pointing to an experimental artifact rather than linkage specificity? Finally, UB-clipping shows that only ~10% of ubiquitin carries two di-glycine remnants, further supporting the idea that HECTD3 primarily mediates attachment of multiple mono-ubiquitination or short chains rather than that of defined long K11 or K48 polyubiquitin chains. Correct?

4) Interpretation of structural data

The high-resolution structure of the HECTD3-Ub intermediate suggests that ubiquitin K11 contributes to the interface with the C-lobe. However, it is unclear how much of this interaction is perturbed by K0 or K11R mutants. Could assays characterizing ubiquitin chains be blurred by introduced mutations?

5) Domain representation

The topological description of the D3NA and D3NB domains is difficult to follow and not well presented in Figure 4d. The claim that residues 45–156 form “eight beta-sheets” sounds like a massive domain and should be replaced by “eight β -strands”. Secondary structure assignments are not always convincing (e.g. β -strand 4), and not all strands are labeled in the figure. A clearer illustration with D3NA and D3NB colored differently, and consistent color coding with Supplemental Figure S1, would be better.

Reviewer #2

(Remarks to the Author)

Huber, Rittinger et al. present a comprehensive structure-function analysis of HECTD3 E3 ligase. HECTD3 is one of the 28 human E3 ligases that all contain a C-terminal catalytic HECT domain, but differ in their N-terminal substrate binding and regulatory domains. In cells, HECTD3 was shown to ubiquitinate Caspase-9, PARP1, c-MYC and STAT3. The PDB holds over 200 structures of HECT E3s, as complexes, full-length proteins, or domains. But before now, there were no structures of HECTD3.

Most of the work is a standard set of protein quality and E3 ligase test experiments. The researchers apply SEC-MALS to show HECTD3 is mainly monomeric. Use of the UBIQUENT E2scan™ kit showed most auto-ubiquitination with UBE2D2 and UBE2D3. The researchers do not find indication of auto-inhibition from testing reactivity with Ub-PR. There is not a single ubiquitin chain linkage formed in vitro. Low resolution cryo-EM structures of HECTD3 show the HECTD3 N-terminal domain and inverted T- and L-shaped HECT domain. Use of Ub-PR afforded model building for a HECTD3-Ub intermediate. At the end of the works, the researchers test ubiquitination of the proposed substrates. PARP1 was ubiquitinated in their hands. The researchers performed XL-MS and observed cross-links between PARP1 DBD and HECTD3 DOC domain. Cross-links were also observed between PARP1 NTDs and HECTD3 HECT domain.

Summary: The study provides new knowledge of the E3 ligase HECTD3. The paper is very well-written. The figures in the main text are clear and very aesthetically-pleasing. The novelty comes from the focus on HECTD3. There are not many structures of HECT E3 complexes with substrates so there would be potential for novelty if the authors added a cryo-EM structure of the complex with PARP1. Otherwise, the work mainly extends to HECTD3 what is known from a large body of work on other HECT E3s. Below are specific points to be addressed in order of Figures.

Figure 1. The HECTD3 E2 preference for UBE2D family members is not unique. Many papers in the last years have shown TRIP12 and UBR5 use UBE2D family members.

Figure 2. It is unclear from the experiments if HECTD3 is being tested for a preferred activity. It is not clear what percentage of the products are monoubiquitin additions and what percentage are ubiquitin chains. It is also unclear if the auto-ubiquitination of the HECT domain alone is on the same site as FL HECTD3. It is also unclear if this HECT domain reaction proceeds in cis or in trans. Since the researchers cannot conclude if the K11R or K48R mutations have effects beyond linkage they should perform western blotting with linkage-specific antibodies if they wish to conclude preferences. Otherwise, the authors should qualify their conclusions about any linkage specificity for HECTD3.

Figure 3. The researchers should consider removing “highly” from the label since the DOC domain is visible in the maps. It is not clear that data support the conclusion that the DOC domain is “highly” flexible compared to the other domains.

Figure 5e and f. Since the mutations affect E3 loading, the researchers cannot rule out that over multiple cycles the impaired activity would not contribute a lot to the effects on auto-ubiquitination. The conclusions about mechanism cannot be drawn without performing Michaelis-Menten enzyme kinetics and pKa measurements. Those experiments would strengthen the conclusions, but they will not affect the novelty of the paper. So the authors could simply qualify their conclusions.

Figure 6. It is difficult to compare PARP ubiquitination and HECTD3 auto-ubiquitination in this gel system. The bands are too compressed. It would be better to separate by a different gel system or analysis by western blotting.

Panel e. The DOC domain Y252A and F256A mutants look like they have general problems. The band below the E2 is lost.

There is not FL auto-ubiquitination even at 60 minutes. The authors need experiments showing that the effects are only on substrate binding. The authors should show the cryo-EM data for these residues as an indicator of model accuracy since the interaction with PARP1 is the most novel part of the work.

Supplementary Figure S1. The statement HECTD3 is highly conserved across species should be reconsidered. The sequence alignment only shows vertebrate species.

Supplementary Figure S2.

The researchers clearly demonstrate that HECTD3 is active with UBE2D family members. They should temper their conclusions however. Studying auto-ubiquitination at 60 minutes for the E2 scan could be misleading because there are a lot of steps other than transthiolation occurring by this point in the experiment.

The – and + labels in panel d are confusing. These look to refer to the E2 added but they probably refer to HECTD3. HECTD3 needs to be added to that line.

Supplementary Figure S3. Is it difficult to see HECTD3 auto-ubiquitination in this gel system. The bands are compressed so it is difficult to differentiate effects of the lysine mutants.

Supplementary Figure S10. The distances to the ubiquitination sites need to be shown if this is to support accuracy of the model.

Supplementary Figure S11. This figure could be more clear. More depthcue or use of surface representation could be help.

Version 1:

Reviewer comments:

Reviewer #1

(Remarks to the Author)

The authorfs did address all my concerns. Congratulations on this lovely piece of work.

Reviewer #2

(Remarks to the Author)

This is a solid, clearly written manuscript with well-presented figures. It is a solid addition to field, with modest conceptual novelty but the first such data for HECTD3. The new experiments in the revision address many of the reviewers' earlier concerns. The remaining points could be addressed with experiments noted below, or, alternatively, by clarifying the limitations of the experiments and conclusions in the text.

Regarding chain branching, the data indicate that HECTD3 alone does not generate branched chains from ubiquitin monomers. However, the possibility that HECTD3 could elaborate or branch chains initiated by another E3 has not been examined. This could be addressed by assays using pre-assembled ubiquitin chains as substrates to test for branching by HECTD3. If those experiments are not feasible at present, revising the text to clearly state this limitation and to acknowledge that future work may reveal a context-dependent role for HECTD3 in chain branching would be necessary.

For the D539A and D855A mutants (Figs. 4e,f after the removal of an original figure, and Supp. Fig. 8d), the data do not unambiguously localize the defect to lysine transfer. Given the observed, potentially pH-sensitive transthiolation defect, the current experiments cannot exclude impairment at the first step to conclude that effects are related to target lysine positioning and deprotonation influenced by acidic residues in the active site. If additional experiments are not feasible, tempering the claims to acknowledge these alternative interpretations would keep the conclusions well aligned with the data.

Response to reviewers

We thank the reviewers for their careful evaluation of our manuscript and their insightful comments and suggestions. To respond to the concerns raised we have carried out additional experiments and made significant changes to the manuscript. These changes include a re-evaluation of the chain linkage specificity detected, more detailed analysis of the DOC domain mutants, analysis of the autoubiquitination sites on HECTD3, investigation into cis or trans ubiquitination and a more detailed analysis of HECTD3 activity with UBE2D versus UBE2L3. To incorporate these changes in a logical manner we have slightly rearranged the sequence of reported results.

We like to thank the reviewers for their suggestions which we believe have significantly improved the manuscript. Please find below detailed answers to individual points raised.

Reviewer #1 (Remarks to the Author):

This manuscript by Huber et al. presents a comprehensive biochemical and structural analysis of HECTD3, a member of the HECT E3 ligase family. While HECT ligases are central regulators of ubiquitin-mediated signaling, their mechanisms of substrate selection and chain specificity remain incompletely understood. HECTD3 has been implicated in inflammation, apoptosis, infection, and cancer progression, yet its structure and enzymatic activity have remained largely uncharacterized. The authors combine biochemical ubiquitination assays, E2 profiling, linkage type analysis, cryo-EM and crystallography, SAXS, XL-MS, and mutational studies to provide the first full-length structural snapshots of HECTD3 in both apo and ubiquitin-loaded states. They reveal a novel N-terminal fold, identify unusual E2 preferences (favoring UBE2D family members over UBE2L3), and present evidence suggesting that HECTD3 catalyzes mainly K11- and K48-linked ubiquitin chains. The study further validates PARP1 as a substrate, maps ubiquitination sites, and proposes mechanistic roles for specific residues in chain-type determination and substrate binding.

The manuscript delivers a rigorous and technically impressive characterization of HECTD3, combining structural biology, biochemistry, and substrate validation to provide new insights into this poorly understood ligase. The cryo-EM reconstructions of full-length HECTD3, supported by crystallography of the DOC domain and SAXS data, are of particular value, revealing two unique N-terminal folds (D3NA and D3NB) and conformational flexibility between L- and inverted T-states characteristic of HECT ligases. The structural work is highly convincing and, aside from a minor figure presentation issue, excellent. The biochemical analyses, however, raise concerns that require further clarification and more cautious interpretation.

We thank this reviewer for their insightful and supportive comments. Please see below for answers to individual points.

1) PARP1 binding and ubiquitination

The XL-MS data supporting PARP1 interaction with HECTD3 are strong, but the biochemical validation is less convincing.

In Figure 6a, the Coomassie-stained SDS-PAGE shows only a modest shift in PARP1 molecular weight, suggesting attachment of few ubiquitin molecules, whereas the western blot in Supplementary Figure S10c shows distinct higher-molecular-weight. It would be helpful if the authors clarified whether PARP1 is mainly multi-mono-ubiquitinated or polyubiquitinated.

The difference in appearance between the Coomassie stained SDS gel and western blot is because longer ubiquitin chains produce stronger immunostaining due to the presence of more epitopes giving the impression that longer ubiquitin chains dominate. Based on reassessment of our data and additional experiments, we now believe that PARP1 becomes mainly multi-mono-ubiquitinated by HECTD3 with some proportion of shorter K48-linked poly-ubiquitin chains (see Supplementary Figs. 10d and 11a). We now report this finding explicitly in the manuscript. Unfortunately, due to the decreased loading of HECTD3 with the ubiquitin K11R mutant (see Supplementary Fig. 9d and further explanation below) we cannot formally distinguish between multi-mono or polyubiquitination by carrying out ubiquitination experiments with the K0 mutant, which would suffer from the same effect.

Do the DOC domain mutants (Y252A, F256A) abolish this higher-molecular-weight species in western blotting, and do they also impair HECTD3 auto-ubiquitination? Furthermore, can the authors exclude that these mutations interfere with E2–E3 charging?

To investigate the integrity of DOC domain mutants, we carried out auto-ubiquitination assays with HECTD3 WT and mutant forms (Supplementary Fig. 12c). These experiments showed no effect on E3 activity. Similarly, to confirm that the mutations do not interfere with E2-E3 transthiolation, we have performed E3 loading assays with the same constructs and confirmed that both mutants are still able to be loaded efficiently with ubiquitin (Supplementary Fig. 12d). We have also carried out additional PARP1 ubiquitination assays followed by both, Coomassie staining and western blot (Supplementary Fig. 12b) and found that the mutations in the DOC domain reduce the higher molecular weight species, indicative of reduced substrate binding.

Finally, is the unusual preference for UBE2D E2s over UBE2L3 also observed in PARP1 ubiquitination assays?

Yes. We have carried out additional PARP1 ubiquitination assays that clearly show that the E2 preference is conserved during substrate ubiquitination (see new Supplementary Fig. 11b).

2) Reported substrates not ubiquitinated in vitro

The failure to ubiquitinate caspase-8, caspase-9, or STAT3 in vitro is surprising, since these proteins are reported in vivo substrates. Could this reflect a requirement for post-translational modifications or adaptor proteins absent in the assay system? For example, phosphorylation of HECTD3 at Thr157 has been reported as essential for caspase-9 ubiquitination, and MALT1 could act as an adaptor for STAT3 or caspase-8.

This is a very good point and we agree, it could be that specific PTMs are required or adaptor proteins that are not present in our *in vitro* assays. We have now added a sentence to the Discussion to highlight this point.

In addition, Supplementary Figure 10 shows that HECTD3 auto-ubiquitination is absent in the presence of caspase-8 and caspase-9, which is puzzling since both are apparently no suitable substrates competing with the former activity; an observation that requires further explanation.

Thank you for highlighting this point. The reason auto-ubiquitination of HECTD3 was not detected in the substrate ubiquitination assays was due to the lower concentration of E3 used in this assay set up. We have now repeated substrate ubiquitination assays at higher concentrations (2 μ M HECTD3 and 4 μ M Substrates) and longer time-points (Supplementary Figs. 10 c and d). Under these conditions we detect strong HECTD3 auto-ubiquitination, which is absent in the reaction with PARP1, where substrate presence suppresses auto-ubiquitination (see new Supplementary Fig. 10).

Ultimately, the idea that a constitutively active, and thus uncontrolled, E3 ligase is present in the cytosol is to some extent worrisome. Is it possible that an additional layer of regulation that restricts HECTD3 from futile or even detrimental ubiquitination events is present in vivo. Although a verification of such regulatory mechanisms is most likely beyond the scope of this study, the authors might still want to include some of their thoughts in the discussion.

This is a good point. It has been reported in the literature that HECTD3, at least in some cases is regulated at the transcriptional level. The key message from our study is that unlike some HECT family E3s HECTD3 activity is not regulated through autoinhibition or oligomerisation. We agree that we cannot exclude that other mechanisms such as specific PTMs or adaptor proteins play a role in a cellular context. We now comment on this in the Discussion.

3) Ubiquitin chain type

The conclusion that HECTD3 primarily catalyzes K11- and K48-linked chains is not fully convincing. Band patterns in Figure 1b and 1c suggest attachment of only a few ubiquitin molecules rather than chain elongation. In UbiCrest assays, residual ubiquitin signals after Cezanne or OTUB1 treatment could reflect priming ubiquitins rather than minor alternative linkages as the authors suggest. Moreover, OTUB1 treatment generates a strong, apparently monoubiquitinated species that persists even with Cezanne co-treatment, possibly pointing to an experimental artifact rather than linkage specificity?

We thank this reviewer for highlighting potential issues with the chain linkage specificity of HECTD3. This is particularly relevant with respect to point 4): the interpretation of structural data and possible contribution of ubiquitin K11 to the interface with HECTD3.

To address this issue, we have first carried out E2-E3 transthiolation assays with the Ub-K11R mutant to test if this mutation may affect overall activity and hence blur any effect on chain linkage that may exist. These transfer assays show indeed that loading is impaired in the case of the K11R mutant (Supplementary Fig. 9d). It is important to note that this will also affect the use of the K0 mutant that is often used to distinguish between multi-mono and polyubiquitination. In light of these new data, we have reevaluated our interpretation of ubiquitination type and chain linkage specificity and now suggest that HECTD3 mainly acts as a chain initiator forming only short ubiquitin chains.

In light of this observation, we have removed the data of the UbCRest assays.

Finally, UB-clipping shows that only ~10% of ubiquitin carries two di-glycine remnants, further supporting the idea that HECTD3 primarily mediates attachment of multiple mono-ubiquitination or short chains rather than that of defined long K11 or K48 polyubiquitin chains. Correct?

Apologies, if we didn't explain the Ub clipping experiment correctly. A single di-glycine remnant signifies that the ubiquitin was part of an unbranched chain or attached to the substrate as mono-ubiquitin, whereas a two di-glycine remnant on ubiquitin signifies that it is a branching point. We agree, that HECTD3 likely mediates primarily multi-mono-ubiquitination and shorter chains. We now say so explicitly in the manuscript.

4) Interpretation of structural data

The high-resolution structure of the HECTD3–Ub intermediate suggests that ubiquitin K11 contributes to the interface with the C-lobe. However, it is unclear how much of this interaction is perturbed by K0 or K11R mutants. Could assays characterizing ubiquitin chains be blurred by introduced mutations?

This is a very good point. Please see our response to point 3) into which we have discussed this issue.

5) Domain representation

The topological description of the D3NA and D3NB domains is difficult to follow and not well presented in Figure 4d. The claim that residues 45–156 form “eight beta-sheets” sounds like a massive domain and should be replaced by “eight b-strands”. Secondary structure assignments are not always convincing (e.g. β -strand 4), and not all strands are labeled in the figure. A clearer illustration with D3NA and D3NB colored differently, and consistent color coding with Supplemental Figure S1, would be better.

We thank the reviewer for this helpful suggestion and changed the text accordingly and re-made the figure to more clearly represent the topology (this is now Fig. 3c). We have also corrected the number of beta strands, which indeed contains 7 instead of 8 (also corrected in Supplementary Fig. 1).

Reviewer #2 (Remarks to the Author):

Huber, Rittinger et al. present a comprehensive structure-function analysis of HECTD3 E3 ligase. HECTD3 is one of the 28 human E3 ligases that all contain a C-terminal catalytic HECT domain, but differ in their N-terminal substrate binding and regulatory domains. In cells, HECTD3 was shown to ubiquitinate Caspase-9, PARP1, c-MYC and STAT3. The PDB holds over 200 structures of HECT E3s, as complexes, full-length proteins, or domains. But before now, there were no structures of HECTD3.

Most of the work is a standard set of protein quality and E3 ligase test experiments. The researchers apply SEC-MALS to show HECTD3 is mainly monomeric. Use of the UBIQUENT E2scan™ kit showed most auto-ubiquitination with UBE2D2 and UBE2D3. The researchers do not find indication of auto-inhibition from testing reactivity with Ub-PR. There is not a single ubiquitin chain linkage formed in vitro. Low resolution cryo-EM structures of HECTD3 show the HECTD3 N-terminal domain and inverted T- and L-shaped HECT domain. Use of Ub-PR afforded model building for a HECTD3-Ub intermediate. At the end of the works, the researchers test ubiquitination of the proposed substrates. PARP1 was ubiquitinated in their hands. The researchers performed XL-MS and observed cross-links between PARP1 DBD and HECTD3 DOC domain. Cross-links were also observed between PARP1 NTDs and HECTD3 HECT domain.

Summary: The study provides new knowledge of the E3 ligase HECTD3. The paper is very well-written. The figures in the main text are clear and very aesthetically-pleasing. The novelty comes from the focus on HECTD3. There are not many structures of HECT E3 complexes with substrates so there would be potential for novelty if the authors added a cryo-EM structure of the complex with PARP1. Otherwise, the work mainly extends to HECTD3 what is known from a large body of work on other HECT E3s. Below are specific points to be addressed in order of Figures.

Figure 1. The HECTD3 E2 preference for UBE2D family members is not unique. Many papers in the last years have shown TRIP12 and UBR5 use UBE2D family members.

This reviewer is absolutely correct. Other HECT ligases have also been shown to work with UBE2D. We did not intend to claim uniqueness. We only wanted to highlight that it is interesting that the activity of a HECT ligase that functions via an E3~Ub thioester intermediate with the cysteine specific E2 UBE2L3 is so low.

Figure 2. It is unclear from the experiments if HECTD3 is being tested for a preferred activity. It is not clear what percentage of the products are monoubiquitin additions and what percentage are ubiquitin chains. It is also unclear if the auto-ubiquitination of the HECT domain alone is on the same site as FL HECTD3. It is also unclear if this HECT domain reaction proceeds in cis or in trans. Since the researchers cannot conclude if the K11R or K48R mutations have effects beyond linkage they should perform western blotting with linkage-specific antibodies if they wish to conclude preferences. Otherwise, the authors should qualify their conclusions about any linkage specificity for HECTD3.

We thank the reviewer for these helpful suggestions and have carried out new experiments to respond to these points.

Here are our answers to individual points raised:

It is not clear what percentage of the products are monoubiquitin additions and what percentage are ubiquitin chains.

Unfortunately, it is not possible to experimentally determine the precise percentage of mono-ubiquitin additions versus poly ubiquitin chains. This is made even harder, given that our new experiments showed that Ub-K11 is important for E2-E3 transthiolation and hence that the ubiquitin K0 mutant cannot be used to assess mono-ubiquitination versus ubiquitin chain formation (please also see our response to reviewer #1, point 3)). However, based on our new experiments we now qualify the text and say that we believe that HECTD3 mainly adds multi-monoubiquitin plus short ubiquitin chains.

It is also unclear if the auto-ubiquitination of the HECT domain alone is on the same site as FL HECTD3.

We have analysed the sites of auto-ubiquitination by mass spec and found that ubiquitination sites in the HECT domain and full-length construct are identical, with additional sites detected in the N-terminal regions of the full-length protein. We prefer not to add these new data to the manuscript as we think that would interrupt the flow of the text.

It is also unclear if this HECT domain reaction proceeds in cis or in trans.

To analyse whether auto-ubiquitination reaction occurs in cis or trans, we performed the following assays: We mixed catalytically inactive GST-tagged HECTD3 HECT domain with wild-type HECT domain and analysed if the former is modified. We found no evidence for modification in trans, even when using western blot detection (anti-Ub antibody). We also performed the assay in the reciprocal manner and similarly detected no convincing sign of trans-ubiquitination of both HECTD3 variants. Nevertheless, given HECTD3's low tendency for oligomerisation, we cannot fully exclude trans-ubiquitination in a cellular complex where other components may contribute. These experiments are shown below. However, as we do not believe that they are important for the overall message of our paper, we would prefer not to add them to the manuscript.

Figure 3. The researchers should consider removing “highly” from the label since the DOC domain is visible in the maps. It is not clear that data support the conclusion that the DOC domain is “highly” flexible compared to the other domains.

We have corrected Figure 3b (now Fig.2b) as requested.

Figure 5e and f. Since the mutations affect E3 loading, the researchers cannot rule out that over multiple cycles the impaired activity would not contribute a lot to the effects on auto-ubiquitination. The conclusions about mechanism cannot be drawn without performing Michaelis-Menten enzyme kinetics and pKa measurements. Those experiments would strengthen the conclusions, but they will not affect the novelty of the paper. So the authors could simply qualify their conclusions.

We do not fully understand this point, particularly in relation to pKa measurements. These mutants were generated to understand if their effect was on the E2-E3 transthiolation step or during the subsequent transfer of ubiquitin on to the substrate, the aminolysis step, which requires a catalytic base to deprotonate the target lysine. Given that we assessed which step is affected for each mutant, it remains unclear to us which mechanistic conclusions should be reconsidered.

Figure 6. It is difficult to compare PARP ubiquitination and HECTD3 auto-ubiquitination in this gel system. The bands are too compressed. It would be better to separate by a different gel system or analysis by western blotting.

We tested several gel systems to improve separation but none provided a meaningful increase in resolution. However, we believe that the separation achieved is sufficient to demonstrate the absence of any modification on HECTD3 in the presence of PARP1 (please see Supplementary Fig. 10d).

Panel e. The DOC domain Y252A and F256A mutants look like they have general problems. The band below the E2 is lost. There is not FL auto-ubiquitination even at 60 minutes. The authors need experiments showing that the effects are only on substrate binding. The authors should show the cryo-EM data for these residues as an indicator of model accuracy since the interaction with PARP1 is the most novel part of the work.

Thank you for raising this point. To ensure that the DOC domain mutations do not affect the overall stability and activity of HECTD3, we have carried out autoubiquitination and E2-E3 transthiolation assays with the mutants. These new experiments are now shown in Supplementary Fig. 12c and d. These experiments show no effect on autoubiquitination, which however is absent when the correct substrate, in this case PARP1, is present (see Supplementary Fig. 10d).

We designed the potential interface DOC domain mutants based on sequence conservation and their location on an exposed loop, likely to be accessible to the substrate. They also lie within the proposed substrate binding region of the APC/C DOC domain. These mutants were not designed based on structural observations, and in fact the side-chain density for these residues is poorly defined in the EM map.

Supplementary Figure S1. The statement HECTD3 is highly conserved across species should be reconsidered. The sequence alignment only shows vertebrate species.

We thank the reviewer for this over-sight have and have corrected the text as suggested.

Supplementary Figure S2.

The researchers clearly demonstrate that HECTD3 is active with UBE2D family members. They should temper their conclusions however. Studying auto-ubiquitination at 60 minutes for the E2 scan could be misleading because there are a lot of steps other than transthiolation occurring by this point in the experiment.

We have carried out the E2 scan according to the manufacturer's instructions. However, to further investigate this point, we have carried out E2-E3 transthiolation assays with UBE2D1 versus UBE2L3 (Supplementary Fig. 2d). These assays clearly show that neither the isolated HECT domain nor full length HECTD3 can efficiently accept ubiquitin from UBE2L3. We hope this resolves any concerns regarding the E2 scan.

The – and + labels in panel d are confusing. These look to refer to the E2 added but they probably refer to HECTD3. HECTD3 needs to be added to that line.

We thank the reviewer for this comment and have modified the figure accordingly (See new Supplementary Fig. 2e)

Supplementary Figure S3. Is it difficult to see HECTD3 auto-ubiquitination in this gel system. The bands are compressed so it is difficult to differentiate effects of the lysine mutants.

We agree and therefore have repeated the experiment over a longer time course, which shows the differences between the mutants more clearly. As noted above, we were unfortunately unable to identify a gel system that provided improved band resolution. Moreover, our experiments with the K11R mutant showed that this mutant is not suitable for assessing chain linkage specificity, as disrupting Ub K11 affects ubiquitin loading due to contacts with HECTD3 (please see also response to reviewer #1, point 3). Nevertheless, we believe it was important to retain these experiments in the manuscript in order to highlight more generally that these mutants are not always suitable for chain type analysis.

Supplementary Figure S10. The distances to the ubiquitination sites need to be shown if this is to support accuracy of the model.

As we have stated in the manuscript on page 11, the predicted HECTD3-PARP1 complex model has low confidence, as indicated by the PAE plot and therefore cannot be used to infer distances to ubiquitination sites. We now recognise that including this model in the manuscript may have given the wrong impression regarding its usefulness and reliability, and have therefore decided to remove it.

Supplementary Figure S11. This figure could be more clear. More depthcue or use of surface representation could be help.

Thank you for this suggestion, we have modified the figure (see new Supplementary Fig. 12e) to better highlight the steric clash at the interface.

Response to remaining comments from reviewer #2:

Reviewer #2 (Remarks to the Author):

This is a solid, clearly written manuscript with well-presented figures. It is a solid addition to field, with modest conceptual novelty but the first such data for HECTD3. The new experiments in the revision address many of the reviewers' earlier concerns. The remaining points could be addressed with experiments noted below, or, alternatively, by clarifying the limitations of the experiments and conclusions in the text.

Regarding chain branching, the data indicate that HECTD3 alone does not generate branched chains from ubiquitin monomers. However, the possibility that HECTD3 could elaborate or branch chains initiated by another E3 has not been examined. This could be addressed by assays using pre-assembled ubiquitin chains as substrates to test for branching by HECTD3. If those experiments are not feasible at present, revising the text to clearly state this limitation and to acknowledge that future work may reveal a context-dependent role for HECTD3 in chain branching would be necessary.

To respond to this point we added the following sentence to the Discussion on page 15: "Furthermore, we can't exclude that HECTD3 might be able itself to branch chains initiated by other E3s."

For the D539A and D855A mutants (Figs. 4e,f after the removal of an original figure, and Supp. Fig. 8d), the data do not unambiguously localize the defect to lysine transfer. Given the observed, potentially pH-sensitive transthiolation defect, the current experiments cannot exclude impairment at the first step to conclude that effects are related to target lysine positioning and deprotonation influenced by acidic residues in the active site. If additional experiments are not feasible, tempering the claims to acknowledge these alternative interpretations would keep the conclusions well aligned with the data.

We have investigated the effect of the D539A and D855A mutants on the first step, the transthiolation step (see Fig. 4e) which showed that this step is reduced by the mutations but not to the same extent that auto-ubiquitination is reduced (see Fig. 4f), supporting the notion that these mutants affect target lysine deprotonation. This interpretation is further supported by the pH-dependent rescue experiments shown in Supplementary Fig.8d.

Given these data we're not entirely sure we understand the specific concern by this reviewer.

In any case, we have been fairly cautious in the interpretation of our data by stating that: "Thus, both residues likely contribute to catalysis, with optimal target lysine positioning and deprotonation depending on the interplay of different acidic side chains in the active site."

Furthermore, we have changed the sentence in the Discussion page 14 "Rather, acidic residues (D539 and D855) of the N-and the C-lobe close to the active site both play a role in lysine de-protonation."

To

“Rather, acidic residues (D539 and D855) of the N-and the C-lobe close to the active site both likely contribute to lysine de-protonation”.